# Effective Targeted Attacks for Adversarial Self-Supervised Learning

**Minseon Kim**[1]**, Hyeonjeong Ha**[1]**, Sooel Son**[1]**, Sung Ju Hwang**[1,2]
[1]Korea Advanced Institute of Science and Technology (KAIST), [2]DeepAuto.ai
{minseonkim, hyeonjeongha, sl.son, sjhwang82}@kaist.ac.kr

## Abstract

Recently, unsupervised adversarial training (AT) has been highlighted as a means of achieving robustness in models without any label information. Previous studies in unsupervised AT have mostly focused on implementing self-supervised learning (SSL) frameworks, which maximize the instance-wise classification loss to generate adversarial examples. However, we observe that simply maximizing the self-supervised training loss with an untargeted adversarial attack often results in generating ineffective adversaries that may not help improve the robustness of the trained model, especially for non-contrastive SSL frameworks without negative examples. To tackle this problem, we propose a novel positive mining for targeted adversarial attack to generate effective adversaries for adversarial SSL frameworks. Specifically, we introduce an algorithm that selects the most confusing yet similar target example for a given instance based on entropy and similarity, and subsequently perturbs the given instance towards the selected target. Our method demonstrates significant enhancements in robustness when applied to non-contrastive SSL frameworks, and less but consistent robustness improvements with contrastive SSL frameworks, on the benchmark datasets.

## 1 Introduction

Enhancing the robustness of deep neural networks (DNN) remains a crucial challenge for their real-world safety-critical applications, such as autonomous driving. DNNs have been shown to be vulnerable to various forms of attacks, such as imperceptible perturbations [14], various types of image corruptions [20], and distribution shifts [25], which can lead DNNs to make incorrect predictions. Many prior studies have proposed using supervised adversarial training (AT) [29, 40, 38, 37] to mitigate susceptibility to imperceptible adversarial perturbation, exploiting class label information to generate adversarial examples. However, achieving robustness in the absence of labeled information has been relatively understudied, despite the recent successes of self-supervised learning across various domains and tasks.

Recently, self-supervised learning (SSL) frameworks have been proposed to obtain transferable visual representations by learning the similarity and differences between instances of augmented training data. Such prior approaches include those utilizing contrastive learning between positive and negative pairs (e.g., Chen et al. [6] (SimCLR), He et al. [19] (MoCo), Zbontar et al. [39] (Barlow-twins)), as well as those utilizing similarity loss solely between positive pairs (e.g., Grill et al. [17] (BYOL), Chen and He [7] (SimSiam)). To achieve robustness in these frameworks, Kim et al. [23] and Jiang et al. [22] have proposed adversarial SSL methods using contrastive learning [6], which generate adversarial examples that maximize the instance-wise classification loss.

Unfortunately, deploying this contrastive framework often becomes computationally expensive as it requires a large batch size for training in order to attain a high level of performance [6]. Specifically, when a given memory and computational budget is limited, such as with edge devices, performing

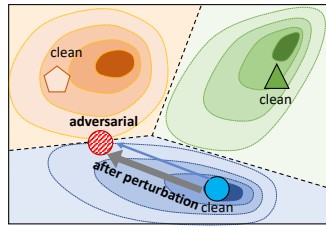 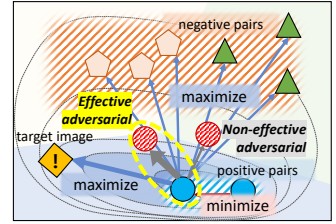 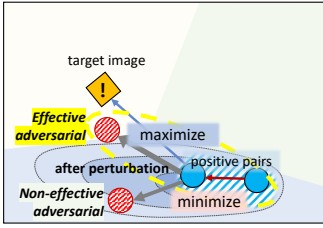

(a) Supervised attack      (b) Contrastive based attack      (c) Positive-pair only attack

Figure 1: **Motivation.** In supervised adversarial learning (a), perturbation is generated to maximize the cross-entropy loss, which pushes adversarial examples to the decision boundaries of other classes. In adversarial contrastive SSL (b), perturbation is generated to minimize the similarity (red line) between positive pairs while maximizing the similarity (blue lines) between negative pairs. In positive-only adversarial SSL (c), minimizing the similarity (red) between positive pairs. However, adversarial examples in adversarial SSL impose weaker constraints in generating effective adversarial examples than does supervised AT due to ineffective positive pairs. To overcome this limitation, we suggest a selectively targeted attack for SSL that maximizes the similarity (blue) to the most confusing target instance (yellow oval in (b) and (c)).

contrastive SSL becomes no longer viable or practical as an option, as it may not obtain sufficiently high performance using a small batch size.

Alternatively, non-contrastive, positive-only SSL frameworks have been proposed resort to maximizing consistency across two differently augmented samples of the same instance, i.e., positive pairs, [17, 7, 39], without the need of negative instances. These approaches improve the practicality of SSL for those limited computational budget scenarios. However, leveraging prior adversarial attacks that maximize the self-supervised learning loss in these frameworks results in extremely poor performance compared to those of adversarial contrastive SSL methods (Table 1). The suboptimality of the deployed attacks causes to learn limited robustness and leads to the generation of ineffective adversarial examples, which fail to improve robustness in the SSL frameworks trained using them. As shown in Figure 1c, the attack in the inner loop of the adversarial training loss, designed to maximize the distance between two differently augmented samples, perturbs a given example to a random position in the latent space. Thus, the generated adversarial samples have little impact on the final robustness. The suboptimality of the attacks also can be occurred in contrastive adversarial SSL that also contains positive pairs, when simply maximizing the contrastive loss. As shown in Figure 1b, contrastive learning treats all positive and negative pairs equally regardless of their varying importance in generating effective adversarial examples.

To address this issue, we propose **T**argeted **A**ttack for **RO**bust self-supervised learning (TARO). TARO is designed to guide the generation of effective adversarial examples by conducting **targeted attacks** that perturb a given instance toward a target instance to enhance the robustness of an SSL framework (Figure 1). The direction of attacks is assigned using our target selection algorithm that chooses the most confusing yet similar sample for a given instance based on the entropy and similarity. By targeting the attacks toward specific latent spaces that are more likely to improve robustness on positive-pairs, TARO improves the robustness of SSL, regardless of the underlying SSL frameworks. Notably, as the positive-pair only SSL has gained attention in recent times, our proposed method becomes crucial for the ongoing safe utilization of these frameworks in real-world applications.

The main contributions can be summarized as follows:

- We observe that simply maximizing the training loss of self-supervised learning (SSL) may lead to suboptimality of attacks as the main cause of the limited robustness in SSL frameworks, especially those that rely on maximizing the similarity between the single pair of augmented instances.

- To address this issue, we propose a novel approach, **T**argeted **A**ttack for **RO**bust self-supervised learning (TARO), which aims to improve the robustness of SSL by conducting targeted attacks on the positive-pair that perturb the given instance toward the most confusing yet similar latent space, based on entropy and similarity of the latent vectors.

- We experimentally show that TARO is able to obtain consistently improved robustness of SSL, regardless of underlying SSL frameworks, including contrastive- and positive-pair only SSL frameworks.

## 2 Related Work

**Adversarial training**    Szegedy et al. [34] showed that imperceptible perturbation to a given input image may lead a DNN model to misclassify the input into a false label, demonstrating the vulnerability of DNN models to adversarial attacks. Goodfellow et al. [14] proposed the fast gradient sign method (FGSM), which perturbs a given input to add imperceptible noise in the gradient direction of decreasing the loss of a target model. They also demonstrated that training a DNN model over perturbed as well as clean samples improves the robustness of the model against FGSM attacks. Follow-up works [27, 2] proposed diverse gradient-based strong attacks, and Madry et al. [29] proposed a projected gradient descent (PGD) attack and a robust training algorithm leveraging a minimax formulation; they find an adversarial example that achieves a high loss while minimizing the adversarial loss across given data points. TRADES [40] proposed minimizing the Kullback-Leibler divergence (KLD) over clean examples and their adversarial counterparts, thus enforcing consistency between their predictions. Recently, leveraging additional unlabeled data [3] and conducting additional attacks [38] have been proposed. Carmon et al. [3] proposed using Tiny ImageNet [28] images as pseudo labels, and Gowal et al. [16] proposed using generated images from generative models to learn richer representations with additional data.

**Self-supervised learning**    Due to the high annotation cost of labeling data, SSL has gained a wide attention [11, 41, 35, 36]. Previously, SSL focused on solving a pre-task problem of collaterally obtaining visual representation, such as solving a jigsaw puzzle [30], predicting the relative position of two regions [11], or impainting a masked area [31]. However, more recently, SSL has shifted to utilizing inductive bias to learn the invariant visual representation of paired transformed images. This is accomplished through contrastive learning, which utilizes both positive pairs and negative pairs, that is differently transformed images and other images from the same batch, respectively [6, 19]. Additionally, some studies have proposed using only positive pairs in SSL and have employed techniques such as momentum networks [17] or stop-gradient [7]. In this paper, we annotate these approaches as contrastive SSL, and positive-pair only SSL, respectively.

**Adversarial self-supervised learning**    The early stage of adversarial SSL methods [23, 22] employed contrastive learning to achieve a high level of robustness without any class labels. Adversarial self-supervised contrastive learning [23, 22] generated an instance-wise adversarial example that maximizes the contrastive loss against its positive and negative samples by conducting untargeted attacks. Both methods achieved robustness, but at the cost of requiring high computation power due to the large batch size needed for contrastive learning. On the other hand, Gowal et al. [15] utilized only positive samples to obtain adversarial examples by maximizing the similarity loss between the latent vectors from the online and target networks, allowing this method greater freedom regarding the batch size. However, it exhibited relatively worse robustness than the adversarial self-supervised contrastive learning frameworks. Despite the advances in the SSL framework (i.e., positive-pair only SSL), a simple combination of untargeted adversarial learning and advanced SSL does not guarantee robustness. To overcome such a vulnerability in positive-pair only SSL, we propose a targeted attack leveraging a novel score function designed to improve robustness.

## 3 Positive-Pair Targeted Attack in Adversarial Self-Supervised Learning

Adversarial SSL and supervised adversarial learning utilize adversarial examples in a similar manner. Specifically, adversarial SSL generates instance-wise adversarial examples in the direction of maximizing the training loss for better robustness. However, this approach exhibits an insufficient level of robustness especially in the positive-pair only self-supervised learning framework due to generating highly suboptimal adversarial examples.

We argue that simply maximizing the training loss, dubbed as an untargeted attack, in positive-pair only SSL limits the diversity of adversarial examples which eventually leads to limited robustness. We theoretically show that range of perturbation is smaller when the positive-pair only SSL objective is employed in an untargeted attack than the contrastive objective in simple two-class tasks. Furthermore, we empirically demonstrate poorer robustness when we naively merge untargeted attack and positive-pair only SSL approaches [17, 7], compared to contrastive-based adversarial SSL [23, 22].

To remedy such a shortcoming, we propose a simple yet effective targeted adversarial attack to increase the diversity of the generated attack. Moreover, we empirically suggest novel positive mining the target for the targeted adversarial attack that contributes to generating more effective and stronger adversarial examples, thus improving the robustness beyond that of previous adversarial SSL approaches. In this section, we first recap supervised adversarial training, self-supervised learning, and previous adversarial SSL methods. We then demonstrate theoretical intuition on our motivation and describe our proposed targeted adversarial SSL framework, TARO, in detail.

## 3.1 Preliminary

**Supervised adversarial training** We first recap supervised adversarial training with our notations. We denote the dataset $\mathcal{D} = \{(x_i, y_i)\}$, where $x_i \in R^D$ is a input, and $y_i \in R^N$ is its corresponding label from the $N$ classes. In this supervised learning task, the model is $f_\theta : X \to Y$, where $\theta$ is a set of model parameters to train.

Given $\mathcal{D}$ and $f_\theta$, an *adversarial attack* perturbs a given source image that maximizes the loss within a certain radius from it (e.g., $\ell_\infty$ norm balls). For example, $\ell_\infty$ attack is defined as follows:

$$\delta^{t+1} = \Pi_{B(0,\epsilon)}\Big(\delta^t + \alpha\texttt{sign}\big(\nabla_{\delta^t}\mathcal{L}_{\texttt{CE}}\big(f(\theta, x + \delta^t), y\big)\big)\Big), \tag{1}$$

where $B(0, \epsilon)$ is the $\ell_\infty$ norm-ball of radius $\epsilon$, $\Pi$ is the projection function to the norm-ball, $\alpha$ is the step size of the attacks, and $\texttt{sign}(\cdot)$ is the sign of the vector. Also, $\delta$ represents the perturbations accumulated by $\alpha\texttt{sign}(\cdot)$ over multiple iterations $t$, and $\mathcal{L}_{\texttt{CE}}$ is the cross-entropy loss. In the case of PGD [29], the attack starts from a random point within the epsilon ball and performs $t$ gradient steps, to obtain a perturbed sample. *Adversarial training* (AT) is a straightforward way to improve the robustness of a DNN model; it minimizes the training loss that embeds the adversarial perturbation ($\delta$) in the inner loop (Eq. 1).

**Self-supervised learning** Recent studies on self-supervised learning (SSL) have proposed methods to allow their models to learn invariant features from transformed images, thus learning semantic visual representations that are beneficial for diverse tasks [6, 19, 17, 7, 39]. In this paper, we aim at improving the robustness of the two most popular types of SSL frameworks: positive-pair only SSL (e.g., BYOL, SimSiam) and contrastive SSL (e.g., SimCLR) frameworks.

We start by briefly describing a representative contrastive SSL, SimCLR [6]. SimCLR is designed to maximize the agreement between different augmentations of the same instance in the learned latent space while minimizing the agreement between different instances. Differently augmented examples from the same instance are defined as positive pairs, and all other instances in the same batch are considered negative examples. Then, the training loss of SimCLR is defined as follows:

$$\mathcal{L}_{\texttt{nt-xent}}(x, \{x_{\texttt{pos}}\}, \{x_{\texttt{neg}}\}) := -\log \frac{\sum_{z_p \in \{z_{\texttt{pos}}\}} \exp(\text{sim}(z, z_p)/\tau)}{\sum_{z_p \in \{z_{\texttt{pos}}\}} \exp(\text{sim}(z, z_p)/\tau) + \sum_{z_n \in \{z_{\texttt{neg}}\}} \exp(\text{sim}(z, z_n)/\tau)}, \tag{2}$$

where $z$ is the latent vector of input $x$, $\texttt{pos}$, $\texttt{neg}$ stands for positive pair and negative pairs of $x$, respectively, and $\text{sim}$ denotes the cosine similarity function.

A representative positive-pair only SSL framework is SimSiam [7]. SimSiam consists of the encoder $f$, followed by the projector $g$, and then the predictor $h$; both $g$ and $h$ are multi-layer perceptrons (MLPs). Given the dataset $\mathcal{D} = \{X\}$ and the transformation function $\mathbf{t} \sim \mathbf{T}$ that augments the images $x \in X$, it is designed to maximize the similarity between the differently transformed images and avoid representational collapse by applying the stop-gradient operation to one of the transformed images as follows:

$$\mathcal{L}_{\texttt{ss}}(x, x_{\texttt{pos}}) = -\frac{1}{2}\frac{p}{||p||_2} \cdot \frac{z_{\texttt{pos}}}{||z_{\texttt{pos}}||_2} - \frac{1}{2}\frac{p_{\texttt{pos}}}{||p_{\texttt{pos}}||_2} \cdot \frac{z}{||z||_2}, \tag{3}$$

where $z = g \circ f(\mathbf{t}_1(x))$, $z_{\texttt{pos}} = g \circ f(\mathbf{t}_2(x))$, and $p = h \circ z$, $p = h \circ z_{\texttt{pos}}$ are output vectors of the projector $g$ and predictor $h$, respectively. Before calculating the loss, SimSiam detaches the gradient on the $z$, which is called the *stop-gradient* operation. This stop-gradient operation helps the model prevent representational collapse without any momentum networks, by making an encoder to act as a momentum network.

**Adversarial SSL** To achieve robustness in SSL frameworks, prior studies have proposed adversarial SSL methods [22, 23, 15]. They generate adversarial examples by maximizing the training loss, dubbed as an untargeted attack, of their base SSL frameworks. For example, the inner loop of an adversarial attack for Kim et al. [23] is structured as follows:

$$\delta^{t+1} = \Pi_{B(0,\epsilon)}\Big(\delta^t + \alpha\,\texttt{sign}\Big(\nabla_{\delta^t}\mathcal{L}\big(\mathbf{t}_1(x) + \delta^t, \mathbf{t}_2(x)\big)\Big)\Big), \tag{4}$$

where the perturbation maximizes the $\mathcal{L}$. For adversarial contrastive SSL approaches [22, 23], $\mathcal{L} = \mathcal{L}_{\texttt{nt-xent}}$ is the contrastive loss in Eq. 2, so that adversarial examples are generated to minimize the similarity between positive pairs and maximize the similarity between negative pairs. For the positive-pair only SSL, adversarial examples are generated to maximize the similarity loss, $\mathcal{L} = \mathcal{L}_{\texttt{ss}}$ (Eq. 3), between positive-pairs only. However, as shown in Table 1, positive-pair only SSL results in significantly poor robustness compared to the adversarial contrastive SSL approaches. This is because using the naive training loss function of positive-pair only SSL in the attack hinders the generation of effective attack images for robust representation, as we theoretically show the range of perturbations is smaller (Section 3.2). To address this issue, we propose a targeted adversarial attack that can select more effective examples to make more diverse perturbations.

## 3.2 Theoretical Motivation: Adversarial Perturbations in Positive-only SSL

A model is considered to have a better generalization of adversarial robustness when the model can maintain its performance across a wide range of adversarial perturbations. Hence, the ability of the attack loss to generate a diverse range of perturbations during training is a crucial factor that influences the model's final robust generalization.

Table 1: Comparison of different attack losses on CIFAR-10 using PGD attack.

| Attack loss | Method | Clean | PGD |
|---|---|---|---|
| Contrastive | ACL [22] | 79.96 | 39.37 |
| | RoCL [23] | 78.14 | 42.89 |
| Positive-only similarity | BYORL [15] | 72.65 | 16.20 |
| | SimSiam* | 71.78 | 32.28 |

*naïve adversarial training applied in SimSiam

However, we found the theoretical motivation that positive-pair only SSL loss ($\mathcal{L}_{\texttt{ss}}$) could not provide a wide range of adversarial perturbations as contrastive loss ($\mathcal{L}_{\texttt{nt-xent}}$) does. We simplify the problem into simple binary classification with the linear layer model to demonstrate our theoretical motivation. Let us denote adversarial perturbations that are generated with both losses as follows,

$$x_{\texttt{ss}}^{\texttt{adv}} = x + \arg\max_{\delta}\left\{\frac{f(x+\delta)}{\|f(x+\delta)\|} \cdot \frac{f(x)}{\|f(x)\|}\right\} \quad \text{subject to} \quad \|\delta\| \leq \epsilon,$$

$$x_{\texttt{nt-xent}}^{\texttt{adv}} = x + \arg\max_{\delta}\left\{-\log\frac{\left(\exp\left(\frac{f(x+\delta)}{\|f(x+\delta)\|} \cdot \frac{f(x)}{\|f(x)\|}/\tau\right)\right)}{\sum\exp\left(\frac{f(x+\delta)}{\|f(x+\delta)\|} \cdot \frac{f(x_{\texttt{neg}})}{\|f(x_{\texttt{neg}})\|}/\tau\right)}\right\} \quad \text{subject to} \quad \|\delta\| \leq \epsilon \tag{5}$$

where we approximate the loss of $\mathcal{L}_{\texttt{ss}}$ in the $\ell_1$ distance function between the positive pair and the loss of $\mathcal{L}_{\texttt{nt-xent}}$ into combination of two $\ell_1$ distance functions of one positive- and one negative- pair. In both cases, a $\delta$ maximizes the respective loss, subject to the constraint that the norm of $\delta$ is less than or equal to $\epsilon$. The objective in positive-only SSL is to make the perturbed and original samples dissimilar as follows,

$$\delta_{\texttt{ss}} = \arg\max_{\delta}|f(x) - f(x+\delta)|. \tag{6}$$

while the objective of nt-xent is to make the perturbed sample dissimilar to the positive pair and similar to the negative pair as follows,

$$\delta_{\texttt{nt-xent}} = \arg\max_{\delta}|f(x) - f(x+\delta)| - |f(x_{neg}) - f(x+\delta)|. \tag{7}$$

**Theorem 3.1** (Perturbation range of self-supervised learning loss). *Given a model trained under the positive-only distance loss, the adversarial perturbations $\delta_{ss}$ are likely to be smaller than those perturbations $\delta_{nt\text{-}xent}$ from a model trained under the positive-pair and negative-pair distance loss. Formally, $\|\delta_{ss}\|_{\infty} < \|\delta_{nt\text{-}xent}\|_{\infty}$.*

These theoretical insights are also supported by the empirical experiments in Table 1 that a model trained with adversarial examples generated using positive- and negative- paired contrastive loss ($\mathcal{L}_{\texttt{nt-xent}}$) have better adversarial robustness generalization because it is exposed to a wider range of perturbations during training than models that are trained with the positive-only similarity loss ($\mathcal{L}_{\texttt{ss}}$). The detailed proof and the empirical analysis are in the Supplementary.

### 3.3 Targeted Adversarial SSL

We propose a simple yet effective targeted adversarial attack to generate effective adversarial examples in a positive-only SSL scenario. In this section, we first show the theoretical intuition of our approach and describe our overall framework to further improve the robustness of the adversarial SSL method by performing targeted attacks wherein targets are selected according to the proposed score function.

**Targeted adversarial attack to different sample** We argue that leveraging untargeted adversarial attacks in positive pairs only SSL still leaves a large room for better robustness. To enlarge the diversity of the attacks, we propose simple targeted adversarial attacks for positive-pair only SSL. The loss for such adversarial attacks is as follows:

$$\delta^{t+1} = \Pi_{B(0,\epsilon)}\Big(\delta^t + \alpha \texttt{sign}\Big(\nabla_{\delta^t}\mathcal{L}_{\texttt{targeted-attack}}\big(x + \delta^t, x')\big)\Big)\Big), \tag{8}$$

where $\mathcal{L}_{\texttt{targeted-attack}}$ is $\mathcal{L}_{\texttt{ours-ss}} = -\mathcal{L}_{\texttt{ss}}$, and $x'$ is a *selected target* within the batch.

Therefore, in the previous simplified scenario described in Section 3.2, conducting the randomly selected targeted attack could increase the range of the perturbation that is generated with positive-pair only similarity loss as follow,

$$\delta_{\texttt{targeted-attack}} = \arg\max_{\delta} |f(x + \delta) - f(x_{\texttt{target}})|. \tag{9}$$

Through triangle inequality, the targeted attack may increase the range of the perturbation and eventually leverage the overall robustness.

**Theorem 3.2** (Perturbation range of targeted attack). *Given a model trained under the $\mathcal{L}_{targeted\text{-}attack}$ loss, the adversarial perturbations $\delta_{targeted\text{-}attack}$ are larger than the adversarial perturbations $\delta_{ss}$ from a model trained under the $\mathcal{L}_{ss}$. Formally, $\|\delta_{targeted\text{-}attack}\|_\infty > \|\delta_{ss}\|_\infty$.*

However, these are theoretical expectations in a simplistic scenario. To further substantiate this, we empirically observed that even a simple targeted attack, with a random target in the batch, significantly improves robustness in a positive-pair only SSL scenario, as shown in Table 2. Therefore, based on these theoretical and empirical insights, we propose to search more effective target for positive-pair targeted attack to boost the robustness of the self-supervised learning frameworks through experimental observations. The detailed proof of Theorem 3.2 is in the Supplementary.

Table 2: Effect of random targeted attack in positive-pair only SSL in CIFAR-5.

| SSL | Attack Type | Clean | PGD |
|---|---|---|---|
| BYOL | untargeted attack | 75.4 | 4.34 |
| | targeted attack | **83.50** | **31.62** |
| SimSiam | untargeted attack* | 66.36 | 36.53 |
| | targeted attack | **77.08** | **47.58** |

*adversarial training applied in SimSiam

**Similarity and entropy-based target selection for targeted attack** In our theoretical analysis and empirical observations, we established that targeted attacks can significantly enhance overall robustness in SSL, except for the target itself. To this end, we propose a score function, denoted as $\mathcal{S}(x, \cdot)$, which aims to identify the most suitable target that is distinct from the input while effectively contributing to improved robustness. Following the studies of Kim et al. [24], Ding et al. [10], Hitaj et al. [21], we prioritize high-entropy examples or those located near decision boundaries as crucial for generating effective adversarial examples in supervised adversarial training. Accordingly, we recommend selecting a target distinct from itself, yet induces confusion, creating adversarial examples that are located close to decision boundaries (Eq.11). The score function yields the most potent target ($x'$) for a given base image ($x$). Subsequently, the targeted attack generates a perturbation, maximizing the similarity to the target $x'$ for the base image $x$.

To this end, we design the score function based on the similarity and entropy values, without using any class information, as follows:

$$\mathcal{S}_{\texttt{entropy}}(x, x') = p'/\tau \log\left(p'/\tau\right), \ \mathcal{S}_{\texttt{similarity}}(x, x') = \frac{e}{|e|_2} \cdot \frac{e'}{|e'|_2}, \tag{10}$$

$$\mathcal{S}_{\texttt{TARO}}(x, x') = \mathcal{S}_{\texttt{entropy}} + \mathcal{S}_{\texttt{similarity}}. \tag{11}$$

where $p = h \circ g \circ f(x)$ and $e = f(x)$ are output vectors of predictor $h$ and encoder $f$, respectively. Overall, the score function $\mathcal{S}$ incorporates both cosine similarity and entropy. The cosine similarity

Table 3: Experimental results against white-box attacks on CIFAR-10. To see the effectiveness, we test TARO on positive-pair only self-supervised learning approaches, i.e., SimSiam, and BYOL.

| Evaluation type | SSL | Attack Type | Clean | PGD | AutoAttack |
|---|---|---|---|---|---|
| Self-supervised linear evaluation | BYOL | $\mathcal{L}_{\texttt{byol}}$ | 72.65 | 16.20 | 0.01 |
| | BYOL | $\mathcal{L}_{\texttt{ours-byol}}$ | **84.52** | **31.20** | **22.01** |
| | SimSiam | $\mathcal{L}_{\texttt{ss}}$ | 71.78 | 32.28 | 24.41 |
| | SimSiam | $\mathcal{L}_{\texttt{ours-ss}}$ | **74.87** | **44.71** | **36.39** |
| Self-supervised robust linear evaluation | BYOL | $\mathcal{L}_{\texttt{byol}}$ | 54.01 | 27.24 | 4.49 |
| | BYOL | $\mathcal{L}_{\texttt{ours-byol}}$ | **74.33** | **40.84** | **29.91** |
| | SimSiam | $\mathcal{L}_{\texttt{ss}}$ | 68.88 | 37.84 | 31.44 |
| | SimSiam | $\mathcal{L}_{\texttt{ours-ss}}$ | **76.19** | **45.57** | **39.25** |

is calculated between features of base images and candidate images in the differently augmented batch. The entropy is calculated with the assumption that the vector $p$ represents the logit of an instance as Caron et al. [4], Kim et al. [24]. Our score function is designed to select an instance $(x')$ that is different but confused with the given image $(x)$, thus facilitating the generation of effective adversarial examples for targeted attack (Figure 1). The experimental results in Figure 2b verify that the score function successfully selects such instances, as intended.

**Robust self-supervised learning with targeted attacks**   The TARO framework starts by selecting a target image based on the score function $(\mathcal{S})$. It then generates adversarial examples using the selected target and performs adversarial training with them.

For a positive pair, represented as differently transformed augmentations $\mathbf{t}_1(x), \mathbf{t}_2(x)$, the target images $\mathbf{t}_2(x')$ and $\mathbf{t}_1(x')$ are selected respectively, as ones with the maximum score within the batch from the score function $(\mathcal{S})$ in Eq. 11. Then, we generate adversarial examples, i.e., $\mathbf{t}_1(x)^{adv}, \mathbf{t}_2(x)^{adv}$, for each transformed input with our proposed targeted attack (Eq. 8), where the targeted loss $\mathcal{L}_{\texttt{targeted-attack}} = -\mathcal{L}_{\texttt{ss}}$ maximizes the similarity to the selected target $\mathbf{t}_2(x')$ and $\mathbf{t}_1(x')$, respectively. Finally, we maximize the agreement between the representations of adversarial images $(\mathbf{t}_1(x)^{adv}$ and $\mathbf{t}_2(x)^{adv})$ and the clean image $t_1(x)$ as follows:

$$\mathcal{L}_{\texttt{TARO}} = \mathcal{L}(\mathbf{t}_1(x), \mathbf{t}_1(x)^{adv}) + \mathcal{L}(\mathbf{t}_1(x)^{adv}, \mathbf{t}_2(x)^{adv}) + \mathcal{L}(\mathbf{t}_2(x)^{adv}, \mathbf{t}_1(x)), \qquad (12)$$

where $\mathcal{L}$ is Eq. 3 for the SimSiam framework. Since all three instances have the same identity, we maximize the similarity between the clean and adversarial examples.

TARO could be also applied to positive pairs in contrastive adversarial SSL methods (e.g., RoCL [23], ACL [22]). Since contrastive SSL does not have a predictor, we use the output of the projector as $p$ in Eq. 10 to select the target for positive-pair. Then, when we apply our targeted attack to their instance-wise attacks, as follows:

$$\mathcal{L}_{\texttt{ours-rocl}} = \mathcal{L}_{\texttt{nt-xent}}(\mathbf{t}_1(x), \{\emptyset\}, \mathbf{t}_1(x)_{\{\texttt{neg}\}}) + \mathcal{L}_{\texttt{similarity}}(\mathbf{t}_1(x), \mathbf{t}_2(x)), \qquad (13)$$

where the adversarial loss is a sum of the modified nt-xent loss [6] and similarity loss. Since TARO alters the untargeted attack of the positive pair with a targeted attack between the base image $(\mathbf{t}_1(x))$ and target image $(\mathbf{t}_1(x'))$, we eliminate the positive pair term in nt-xent loss and add similarity loss instead. The similarity loss maximizes the cosine similarity between the $\mathbf{t}_1(x)$ images and the $\mathbf{t}_1(x')$ images which are searched by the score function. Overall, we generate adversarial examples that maximize the $\mathcal{L}_{\texttt{ours-rocl}}$ loss as shown in Algorithm 1.

## 4   Experiment

In this section, we extensively evaluate the efficacy of TARO with both contrastive and positive-pair only adversarial SSL frameworks. First, we compare the performance of our model to previous adversarial SSL methods that do not utilize any targeted attacks in Section 4.1. Moreover, we evaluate the robustness of the learned representations across different downstream domains in Section 4.2. Finally, we analyze the reason behind the effectiveness of targeted attacks in achieving better robust representations compared to models using untargeted attacks in Section 4.3.

**Experimental setup** We compare TARO against previous contrastive and positive-pair only adversarial SSL approaches. Specifically, we adapt TARO on top of two contrastive adversarial SSL frameworks, RoCL [23], ACL [22] and a positive-pair only SSL framework, SimSiam [7], to demonstrate its efficacy in enhancing their robustness. All models use the ResNet18 backbones that are trained on CIFAR-10 and CIFAR-100 with $\ell_\infty$ PGD attacks with the attack step of 10 and epsilon $8/255$. We evaluate the robustness of our method against two types of attack, AutoAttack* [8] and $\ell_\infty$ PGD attacks, with the epsilon size of $8/255$, using the attack step of 20 iterations. Clean denotes the classification accuracy of the ResNet18 backbone on the original images. We further describe the experimental details in Appendix B. Code is available in https://github.com/Kim-Minseon/TARO.git

## 4.1 Efficacy of Targeted Attacks in Adversarial SSL

We first validate whether the proposed targeted attacks in TARO contribute to improving the robustness of positive-pair adversarial SSL frameworks. To evaluate the quality of the learned representations with the SSL frameworks, we utilize linear and robust linear evaluation, as shown in Table 3. Then, we validate the generality of TARO to contrastive-based adversarial SSL frameworks (Table 5).

**Robustness improvements in positive-pair only SSL**
We evaluate the efficacy of TARO by comparing those to untargeted attacks on positive-pair only SSL frameworks, i.e., SimSiam and BYOL. As shown in Table 3, when replacing untargeted attacks with TARO in the positive-only SSL, TARO contributes to attaining significant gains in both robustness accuracy against PGD attacks and clean accuracy. This is due to the inherent limitations of untargeted attacks in positive-pair only SSL frameworks. In such frameworks, perturbations in any direction away from the other pair of samples will

Table 4: Ablation results on target selection.

| Method | Selection | Clean | PGD |
|--------|-----------|-------|------|
| RoCL | None | 78.14 | 42.89 |
| | Random | 79.26 | 43.45 |
| | Ours | 80.06 | 45.37 |
| SimSiam | None* | 71.78 | 32.28 |
| | Random | 73.25 | 42.85 |
| | Ours | 74.87 | 44.71 |

*naïve adversarial training applied in SimSiam

inevitably increase the SSL loss, making it challenging to generate effective adversarial examples. However, with the guidance provided by TARO, the model is able to generate stronger attack images, leading to meaningfully improved performance both on clean and adversarially perturbed images. Furthermore, we show that the untargeted attacks are not only ineffective for learning robust features, but also hinder the learning of good visual representation for clean images.

Switching from an untargeted to a targeted attack approach leads to a substantial increase in performance across both contrastive-based and positive-pair only approaches, as shown in Table 4. This advancement is particularly evident when addressing the challenge of selecting appropriate targets within positive pairs. As we have discussed in the Limitations section, our empirical score function may not be the absolute optimal algorithm for target selection. Nevertheless, it is clear that concentrating on targeted attacks in the context of positive pairs is crucial for enhancing robust representation, applicable to both clean and adversarial examples.

**Robustness improvements in contrastive adversarial SSL** The robustness gains through TARO in contrastive adversarial SSL, specifically RoCL and ACL, are demonstrated in Table 5. Given that our TARO algorithm mines the positive-pair in contrastive loss, its effects on contrastive-based SSL might be more limited compared to positive-pair SSL. Despite this, TARO enhances RoCL's robustness against PGD attacks from 42.89% to 45.37% without compromising the clean accuracy. In the case of ACL, TARO fortifies the robustness against PGD attacks while maintaining performance comparable to AutoAttack.

## 4.2 Evaluation on CIFAR-100

**Robustness on larger benchmarks datasets** We further validate our method on a larger dataset, CIFAR-100. In Table 6, TARO demonstrates consistent robust accuracy when compared with those of the adversarial SSL frameworks using untargeted attacks, with notably significant robustness improvements on the positive-pair only SSL. Although the clean and original robust accuracy of the positive-only SSL method is noticeably lower than that of the contrastive learning method on this particular dataset, it achieves significantly higher robust accuracy than the contrastive counterpart

---

*https://github.com/fra31/auto-attack

Table 5: Experimental results against white-box attacks on ResNet18 trained on the CIFAR-10 dataset. To see the effectiveness, we test TARO on contrastive adversarial SSL, i.e., RoCL, and ACL.

| Evaluation type | Method | Attack Type | Clean | PGD | AutoAttack |
|---|---|---|---|---|---|
| Self-supervised linear evaluation | RoCL [23] | $\mathcal{L}_{\text{rocl}}$ | 78.14 | 42.89 | 27.19 |
| | +TARO | $\mathcal{L}_{\text{ours-rocl}}$ | **80.06** | **45.37** | **27.95** |
| | ACL [22] | $\mathcal{L}_{\text{acl}}$ | **79.96** | 39.37 | **35.97** |
| | +TARO | $\mathcal{L}_{\text{ours-acl}}$ | 78.45 | **39.71** | 35.81 |

Table 6: Results of linear evaluation in a larger dataset, CIFAR-100.

| Method | Clean | PGD |
|---|---|---|
| RoCL | 45.99 | 17.17 |
| +TARO | **46.54** | **18.91** |
| SimSiam* | 24.43 | 13.34 |
| +TARO | **36.02** | **22.18** |

when using our targeted attack. The results further suggest that the proposed targeted attack plays a crucial role in creating effective adversarial examples.

**Transferable robustness** The main objective of SSL is to learn transferable representations for diverse downstream tasks. Therefore, we further evaluate the transferable robustness of the pretrained representations trained using our targeted attack on novel tasks from a different dataset. We adopt the experimental setting from the previous works on supervised adversarial transfer learning [32] which freeze the encoder and train only the fully connected layer. We pretrained the model on CIFAR-100 and evaluate the robust transferability to CIFAR-10. In Table 7, our model also shows impressive transferable robustness both with contrastive and positive-pair only SSL, compared to those obtained by the representations learned with untargeted adversarial SSL.

Table 7: Results of adversarial transfer learning to CIFAR-10 from CIFAR-100.

| Method | Clean | PGD |
|---|---|---|
| RoCL | **73.93** | 18.62 |
| +TARO | 65.21 | **19.13** |
| SimSiam* | **53.34** | 11.24 |
| +TARO | 50.50 | **25.44** |

## 4.3 Effectiveness of TARO

In this section, we further analyze the effect of the targeted attacks in adversarial SSL to see how and why it works. 1) Analysis of the selected images by $\mathcal{S}$, 2) Visual representation of adversarial examples that are generated with untargeted attack/targeted attack, and 3) ablation experiment on each component of the score function.

**Analysis of the selected target** To analyze which target images are selected by our score function ($\mathcal{S}$), we use a supervised adversarial training (AT) model. We select the target images of a single class (airplane) with the score function, and forward them to the supervised AT model to obtain their class distribution. To further examine which are the most confusing classes for the original images, we forward the base airplane images to the supervised AT model as well. As shown in Figure 2a, airplane images are easily confused with the ship class and the bird class. Surprisingly, 1/3 of the target images are selected using our target selection function for airplane images belonging to either ship or the bird class, which are the most confusing classes for the images belonging to the airplane class (See Figure 2b). These results strongly support that our score function effectively selects targets that are similar yet confused, as intended, without using any label information.

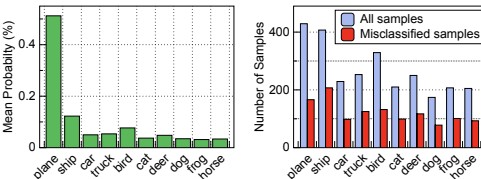

(a) Mean predict probability of base images
(b) Distribution of class of targeted images

Figure 2: Analysis of target from score function ($\mathcal{S}$)

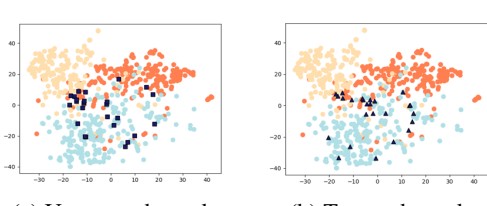

(a) Untargeted attack
(b) Targeted attack

Figure 3: Visualize embedding

**Visualization of embedding space** To examine the differences between images that are generated with targeted and untargeted attacks, we visualize their embedding space. In Figure 3, black markers represent adversarial examples, and light blue markers represent clean examples, both belonging to the same class. As shown in Figure 3a, untargeted adversarial examples are located near clean examples,

---

*naïve adversarial training applied in SimSiam

and far from the class boundaries. On the other hand, targeted adversarial examples are located near the class boundaries (Figure 3b), although it is generated in an unsupervised manner without any access to class labels. This visualization shows that our targeted attack generates relatively more effective adversarial examples than untargeted attacks, which is likely to push the decision boundary to learn more discriminative representation space for instances belonging to different classes.

**Ablation study of the score function** To demonstrate the effect of each component in our score function, we conduct an ablation study of the score function $\mathcal{S}$. The score function consists of two terms: the entropy term and the cosine similarity term (Eq. 10), which together contribute to finding an effective target that is different but confusing. We empirically validate each term by conducting an ablation experiment using only a single term in the score function during adversarial SSL training in Eq. 11. The experimental results in Table 8, suggest that the entropy term leads to good clean accuracy while the similarity term focuses on achieving better robust performance. Thus the combined score function enables our model to achieve good robustness while maintaining its accuracy on clean examples.

Table 8: Results of ablation study on score function on CIFAR-10.

|  | Clean | PGD | AutoAttack |
|---|---|---|---|
| $\mathcal{S}_{\texttt{entropy}}$ | 78.43 | 40.35 | 32.51 |
| $\mathcal{S}_{\texttt{similarity}}$ | 72.90 | 44.59 | 36.12 |
| $\mathcal{S}_{\texttt{TARO}}$ | 74.06 | 44.71 | 36.39 |

## 5   Conclusion

In this paper, we demonstrate that a simple combination of supervised adversarial training with self-supervised learning is highly suboptimal due to the ineffectiveness of adversarial examples generated by untargeted attacks in positive-pair only SSL, which perturb to random latent space without considering decision boundaries. To address this limitation, we proposed an instance-wise targeted attack scheme for adversarial self-supervised learning. This scheme selects the target instance based on similarity and entropy, such that the given instance is perturbed to be similar to the selected target. Our targeted adversarial self-supervised learning yields representations that achieve better robustness when applied to any type of adversarial self-supervised learning, including positive-pair only SSL and contrastive SSL. We believe that our work paves the way for future research in exploring more effective attacks for adversarial self-supervised learning.

## Limitations

Our method's main constraint is that our score function's design relies on empirical design based on the previous works. Establishing the most optimal score function theoretically for a high-dimensional, non-linear deep learning model is a complex task. Despite this, we've provided a theoretical basis for how a targeted attack can improve robustness in a simple scenario for positive pairs. Our experimental results also confirm our score function's effectiveness, suggesting we've made various efforts to counterbalance our limitations. Additionally, our method demands more computational time than a simple untargeted adversarial training, given the need to select a target instance. Yet, this extra computational time is less than 5% compared to original training time. Considering the significant boost in robustness, we believe it's a reasonable trade-off to implement our method. Despite these limitations, we've identified a significant vulnerability in the untargeted attack method—an essential discovery for adversarial self-supervised learning. Moreover, we suggest a simple yet effective way to address this vulnerability in adversarial self-supervised learning.

## Acknowledgement

This work was supported by Institute of Information & communications Technology Planning & Evaluation (IITP) grant funded by the Korea government (MSIT) (No.2020-0-00153) and by Institute of Information & communications Technology Planning & Evaluation (IITP) grant funded by the Korea government(MSIT) (No.2019-0-00075, Artificial Intelligence Graduate School Program(KAIST)). We thank Jin Myung Kwak, Eunji Ko, Jihoon Tack, and Yulmu Kim for providing helpful feedbacks and support in journey of this research. We also thank the anonymous reviewers for their insightful comments and suggestions.

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

# Effective Targeted Attack for
# Adversarial Self-Supervised Learning

## Supplementary Material

## A    Baselines.

- **RoCL [23].** RoCL is SimCLR [6] based adversarial self-supervised learning methods. We experiment with the official code[†]. To make a fair comparison, we set the attack step to 10 as other baselines. We train the model with 1,000 epochs under the LARS optimizer with weight decay $2e-6$ and momentum with 0.9. For the learning rate schedule, we also followed linear warmup with cosine decay scheduling. We set a batch size of 512 for all datasets (CIFAR-10, CIFAR-100, STL10). For data augmentation, we use a random crop with 0.08 to 1.0 size, horizontal flip with a probability of 0.5, color jitter with a probability of 0.8, and grayscale with a probability of 0.2 for RoCL training.

- **ACL [22].** ACL is SimCLR [6] based adversarial self-supervised learning methods. We conduct the experiment with the official code[‡]. To make a fair comparison, we set the attack step to 10 as other baselines. We train the model with 1,000 epochs. We set a batch size of 512 for STL10 dataset. For CIFAR-10, and CIFAR-100, we use the official pretrained checkpoints. For data augmentation, we use a random crop with 0.08 to 1.0 size, horizontal flip with a probability of 0.5, color jitter with a probability of 0.8, and grayscale with a probability of 0.2 for ACL training. We set PGD dual mode which calculates both clean and adversarial during the training.

- **BYORL [15]** BYORL is BYOL [17] based adversarial self-supervised learning methods for low label regime. Since there is no official code for BYORL we implement the BYORL by ourselves. We implement based on BYOL from a self-supervised learning library [§]. We use the same CIFAR-10 setting in the library except for normalization. We exclude normalization in the data augmentation. To make a fair comparison, we implement on the ResNet18 with attack step 10 of PGD. As shown in supplementary materials in [15], when the model is trained with 10 steps in ResNet34 it shows 37.88% of robustness. We conjecture that we have a different performance from the original paper because the original paper employs 40 steps of PGD in WideResNet34 to obtain the reported robustness which requires extraordinary computation power.

- **AdvCL [12].** AdvCL is SimCLR [6] based adversarial self-supervised learning which employ pseudo labels from the model that is pretrained on ImageNet [26] data. Even though the outstanding performance of AdvCL, we exclude this model as our baseline because the proposed methods require the model that is trained with the labels of ImageNet which we assume to have no label information for training.

## B    Detailed description of experimental setups.

### B.1    Resource description.

All experiments are conducted with a two NVIDIA RTX 2080 Ti, except for the experiments with CIFAR-100 experiments. For CIFAR-100 experiments, two NVIDIA RTX 3080 are used. All experiments are processed in Intel(R) Xeon(R) Silver 4114 CPU @ 2.20GHz.

### B.2    Training detail.

For all methods, we train on ResNet18 [18] with $\ell_\infty$ attacks with attack strength of $\epsilon = 8/255$ and step size of $\alpha = 2/255$, with the number of inner maximization iterations set to $K = 10$. For the optimization, we train every model for 800 epochs using the SGD optimizer with the learning rate of

---

[†]https://github.com/Kim-Minseon/RoCL
[‡]https://github.com/VITA-Group/Adversarial-Contrastive-Learning
[§]https://github.com/vturrisi/solo-learn

**Algorithm 1** Targeted Attack Robust Self-Supervised Learning (TARO) for contrastive-based SSL

---

**Input:** Dataset $\mathcal{D}$, transformation function $\mathbf{t}$, model $f$, parameter of model $\theta$, target score function $\mathcal{S}$
**for** iter $\in$ number of iteration **do**
    **for** $x_i \in$ miniBatch $B = \{x_1, \ldots, x_m\}$ **do**
        **for** n in 2 **do**
            Transform input $\mathbf{t}_n(x_i)$
            Find target images $\mathbf{t}_n(x_k)$ from $\mathcal{S}(\mathbf{t}_n(x_i), \text{batch})$
            Generate targeted adversarial examples
            $\mathcal{L}_{\texttt{cont-attack}} = \mathcal{L}_{\texttt{nt-xent}}(\mathbf{t}_1(x_i), \{\emptyset\}, \{\mathbf{t}_1(x_i)_{\{\texttt{neg}\}}\}) + \mathcal{L}_{\texttt{similarity}}(\mathbf{t}_1(x_i), \mathbf{t}_2(x_k))$
            $\delta^{t+1} = \Pi_{B(0,\epsilon)}\left(\delta^t + \alpha\texttt{sign}\left(\nabla_{\delta^t}[\mathcal{L}_{\texttt{cont-attack}}]\right)\right)$
            $\mathbf{t}_n(x_i)^{adv} = \mathbf{t}_n(x_i) + \delta^t$
        **end for**
        Calculate training loss
        $\mathcal{L}_{\texttt{TARO}} = \mathcal{L}_{\texttt{nt-xent}}(\mathbf{t}_1(x_i), \{\mathbf{t}_2(x_i), \mathbf{t}_1(x_i)^{adv}\}, \{\mathbf{t}_1(x_i)_{\{\texttt{neg}\}}\})$
    **end for**
    $\theta \leftarrow \theta - \beta\nabla_\theta\mathcal{L}_{\texttt{TARO}}$
**end for**

---

0.05, weight decay of $5\mathrm{e}{-4}$, and the momentum of 0.9. For data augmentation, we use a random crop with 0.08 to 1.0 size, horizontal flip with a probability of 0.5, color jitter with a probability of 0.8, and grayscale with a probability of 0.2. We exclude normalization for adversarial training. We set the weight of adversarial similarity loss $w$ as 2.0. We use batch size 512 with two GPUs.

In the score function, we calculate the similarity score term and the entropy term as shown in Equation 11. First, to exclude the positive pairs' similarity score we set the similarity score between positive pairs to $-1$. Then, to calculate the overall score, after obtaining the similarity score and entropy of each sample, we normalize each component with Euclidean normalization to balance each component to the score function. Further, the detailed algorithm of TARO for contrastive SSL is described in Algorithm 1 and Eq. 14.

## B.3 Evaluation details.

**PGD $\ell_\infty$ attack.** For all PGD $\ell_\infty$ attacks used in the test time, we use the projected gradient descent (PGD) attack with the strength of $\epsilon = 8/255$, with the step size of $\alpha = 8/2550$, and with the number of inner maximization iteration set to $K = 20$ with the random start.

**AutoAttack.** We further test against a strong gradient-based attack, i.e., AutoAttack (AA) [8]. AutoAttack is an ensemble attack of four different attacks (APGD-CE, APGD-T, FAB-T [9], and Square [1]). AGPD-CE is an untargeted attack, APGD-T and FAB-T are targeted attacks. The Square is a black-box attack. We use an official code to test models[¶].

**Self-supervised learning.** For self-supervised learning, we denote linear evaluation when we use only clean images to train the fully connected (fc) layer after the pretraining phase. When we denote robust linear evaluation, we train the fc layer with adversarial examples. While ACL uses partial fine-tuning to obtain their reported accuracy and robustness, to make a fair comparison, we freeze the encoder and train only the fc layer. Robust fine-tuning is training all parameters including parameters of the encoder with adversarial examples. For linear evaluation, we followed the baseline hyperparameters for each model. We train the baseline models with 150 epochs, 25 epochs, and 50 epochs for RoCL, and ACL, respectively. We also followed their learning rate of 0.1, 0.1, and $2 \times 10^{-3}$ for RoCL, and ACL, respectively. On the other hand, we train our model with 100 epochs with a learning rate of 0.5 for linear evaluation. We use AT loss for robust linear evaluation except for ACL. For ACL, we use TRADES loss as the official code.

## C  Experimental Details of Analysis.

**Analysis the distribution of target class.** To analyze the target from the score function ($\mathcal{S}$), we employ an adversarially supervised trained model. We calculate the score function that is trained with our TARO on SimSiam. We use a train set. For each class, we calculate the mean predict probability, which is the average of all softmax outputs of target images from the supervised trained

---

[¶]https://github.com/fra31/auto-attack

model. Further, we also count the number of samples that are predicted for each class. In Figure 2, the results are target images of the airplane as a base image. There is a similar tendency even though we change the base class to other classes as shown in the following Figure 4.

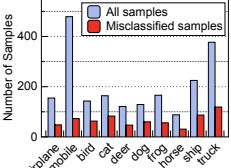

(a) Distribution of class of target of *automobile*

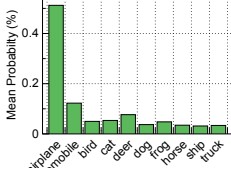

(b) Mean predict probability of *automobile*

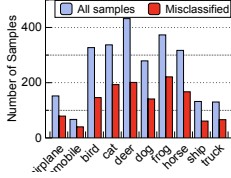

(c) Distribution of class of target of *deer*

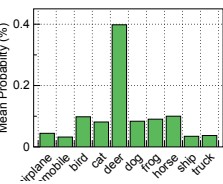

(d) Mean predict probability of *deer*

Figure 4: Analysis of target distribution in different classes

**Visualization of embedding space.** To visualize the embedding of our targeted attack and untargeted attack, we use t-Distributed Stochastic Neighbor Embedding (t-SNE) [5] with the cosine similarity metric. Our TARO model is trained on CIFAR-10 as a feature extractor. We sample a few examples and conduct two types of attack, the untargeted attack and the targeted attack. To visualize more effectively we ignore the other seven classes in CIFAR-10. We visualize clean examples from three classes and then visualize adversaries that are generated with our targeted attack and untargeted attack, respectively, with dark blue.

# D   Additional Experiment

**Contrastive based adversarial self-supervised learning with TARO.**    Our TARO could be also applied to positive pairs in contrastive-based adversarial self-supervised learning (e.g., RoCL [23], ACL [22]). We applied our TARO in instance-wise attack of the contrastive-based approaches as follow,

$$\mathcal{L}_{\texttt{attack}} = \mathcal{L}_{\texttt{nt-xent}}(x, \{\emptyset\}, \{x_{\texttt{neg}}\}) + \mathcal{L}_{\texttt{similarity}}(x, \{x_{j_{\texttt{TARO}}}\}) \qquad (14)$$

where attack loss is consists of original attack loss nt-xent loss [6] and similarity loss. The similarity loss additionally constrains the positive pairs as the TARO that maximize the similarity between the $x$ with the $j^{th}$ index images which is searched by our TARO score function. Overall, we generate adversarial examples that maximizes the $\mathcal{L}_{\texttt{attack}}$ loss. Surprisingly, when we apply TARO on the contrastive learning based approach, previous work could achieve marginally better clean accuracy and robustness. This shows that our empirical assumption also holds on contrastive-based SSL but since there is (1/batch size) effects on the total loss the gain could be marginal.

**Robustness against black box attack**    We conduct black box attack to verify our model is robust to gradient free attacks. We generate black box adversaries with AT [29] model, RoCL [23] model and our models. Then, we test adversaries to each other. As show in the table, our model is able to defend the black box attack from AT model than the RoCL model. Moreover, our model generates stronger black box adversaries than RoCL since AT model shows more weak robustness.

Table 9: Results of black box attack. Models on the row are the tested models. Models on the columns are the source models to generate black box adversaries.

|       | AT    | RoCL  | Ours      |
|-------|-------|-------|-----------|
| AT    | -     | 59.73 | **60.92** |
| RoCL  | 70.40 | -     | 57.98     |
| Ours  | **69.97** | 54.99 | -     |

**Robustness against diverse attacks**    We tested our approach against diverse types of adversarial attacks, including the Carlini-Wagner (CW) attack [2], black-box attack, i.e., Pixle [33], and Patch-attack, i.e., PIFGSM [13], as shown in Table 10. Since our approach already showed improved performance against Autoattack, which includes black-box Square attacks, our approach is able to consistently demonstrates enhanced robustness against both the CW attack and black-box attacks.

Table 10: Results against diverse attacks.

| Method   | PGD   | CW    | Pixle | PIFGSM |
|----------|-------|-------|-------|--------|
| RoCL     | 42.89 | 76.45 | 67.32 | 43.23  |
| +TARO    | 45.37 | 72.75 | 68.40 | 44.56  |
| SimSiam  | 32.28 | 68.14 | 54.56 | 28.31  |
| +TARO    | 44.97 | 73.87 | 67.22 | 46.37  |

# E   Proof of the Theorem

Let us consider the problem as a simple binary task using a linear layer model to demonstrate our theoretical motivation. The dataset $\mathcal{D} = X, \cdot$ consists of training examples, where $x \in X$ represents a training example without any class label. We assume there is a single positive pair and a single negative pair. The linear model is denoted as $f(\cdot)$. The adversarial perturbations generated using both losses are as follows:

$$x_{\text{ss}}^{\text{adv}} = x + \arg\max_{\delta} \left\{ \frac{f(x+\delta)}{\|f(x+\delta)\|} \cdot \frac{f(x)}{\|f(x)\|} \right\} \quad \text{subject to} \quad \|\delta\| \leq \epsilon,$$

$$x_{\text{nt-xent}}^{\text{adv}} = x + \arg\max_{\delta} \left\{ -\log \frac{\left( \exp\left( \frac{f(x+\delta)}{\|f(x+\delta)\|} \cdot \frac{f(x)}{\|f(x)\|} / \tau \right) \right)}{\exp\left( \frac{f(x+\delta)}{\|f(x+\delta)\|} \cdot \frac{f(x_{\text{neg}})}{\|f(x_{\text{neg}})\|} / \tau \right)} \right\} \quad \text{subject to} \quad \|\delta\| \leq \epsilon \tag{15}$$

where we approximate the cosine similarity distance loss into $\ell_1$ distance function. In both cases, a $\delta$ maximizes the respective loss, subject to the constraint that the norm of $\delta$ is less than or equal to $\epsilon$. The objective in positive-only SSL is to make the perturbed and original samples dissimilar as follows,

$$\delta_{\text{ss}} = \arg\max_{\delta} |f(x) - f(x+\delta)|, \tag{16}$$

$$\delta_{\text{nt-xent}} = \arg\max_{\delta} |f(x) - f(x+\delta)| - |f(x_{neg}) - f(x+\delta)|. \tag{17}$$

The range of adversarial attack of each loss is then calculated as follow,

$$\|\delta_{\text{ss}}\| = \| \arg\max_{\delta} |f(x) - f(x+\delta)| \|$$
$$= \arg\max_{\delta} (|f(x) - f(x+\delta)|)^2 \tag{18}$$

$$\|\delta_{\text{nt-xent}}\| = \| \arg\max_{\delta} (|f(x) - f(x+\delta)| - |f(x_{\text{neg}}) - f(x+\delta)|) \|$$
$$= \arg\max_{\delta} \big( |f(x) - f(x+\delta)|^2 - 2|f(x) - f(x+\delta)| \cdot |f(x_{\text{neg}}) - f(x+\delta)|$$
$$+ |f(x_{\text{neg}}) - f(x+\delta)|^2 \big)$$
$$\approx \arg\max_{\delta} \big( |f(x) - f(x+\delta)|^2 - 2|\delta| \cdot |f(x_{\text{neg}}) - f(x+\delta)| + |f(x_{\text{neg}}) - f(x+\delta)|^2 \big)$$
$$\approx \arg\max_{\delta} \big( |f(x) - f(x+\delta)|^2 + |f(x_{\text{neg}}) - f(x+\delta)|^2 \big) \quad \because \delta \leq \epsilon$$
$$\geq \arg\max_{\delta} |f(x) - f(x+\delta)|^2. \tag{19}$$

If there are more negative pairs, the difference in perturbation range between positive-pair-only attacks and contrastive attacks could become more pronounced.

**Theorem E.1** (Perturbation range of self-supervised learning loss). *Given a model trained under the positive-only distance loss, the adversarial perturbations $\delta_{ss}$ are likely to be smaller than those perturbations $\delta_{nt-xent}$ from a model trained under the positive-pair and negative-pair distance loss. Formally, $\|\delta_{ss}\|_{\infty} < \|\delta_{nt-xent}\|_{\infty}$.*

When applying a random targeted attack within the positive-pair-only self-supervised learning framework, we can effectively increase the range of perturbations. Let us assume that the target instance $x_{\text{target}}$ is different from the original instance $x$, and the distance between them is greater than the threshold $\delta$. The perturbations generated through the targeted attack are as follows:

$$\delta_{\text{targeted-attack}} = \arg\max_{\delta} |f(x+\delta) - f(x_{\text{target}})|. \tag{20}$$

Let us denote target instance $x_{\text{target}}$ as $x'$ for simple equations,

$$|f(x) - f(x+\delta)| < |f(x') - f(x+\delta)|$$
$$\because |x' - x| > \delta$$
$$= |f(x+\delta) - f(x')| \tag{21}$$
$$\therefore \| \arg\max_{\delta} |f(x) - f(x+\delta)| \| < \| \arg\max_{\delta} |f(x+\delta) - f(x')| \|$$

The random targeted attack, which targets instances that are at a greater distance than $\delta$ from the original input, can potentially increase the perturbation range and ultimately enhance overall robustness.

**Theorem E.2** (Perturbation range of targeted attack)**.** *Given a model trained under the $\mathcal{L}_{targeted\text{-}attack}$ loss, the adversarial perturbations $\delta_{targeted\text{-}attack}$ are likely to be larger than those from a model trained under the $\mathcal{L}_{ss}$. Formally, $\|\delta_{targeted\text{-}attack}\|_\infty > \|\delta_{ss}\|_\infty$.*

# F    Broader Impacts

The pursuit of adversarial robustness against malicious attacks within deep neural networks remains an unsolved, yet fundamental area of deep learning research. To date, several self-supervised adversarial training approaches have been proposed, primarily based on the contrastive learning framework. However, the attainment of robustness via a 'positive-pair only' self-supervised learning approach is still under-explored. Consequently, self-supervised frameworks have evolved from large batch contrastive learning to a focus on single 'positive-pair only' learning paradigms. The area of self-supervised learning that we are targeting aims to delve into the robustness of these new learning frameworks through our tailored attacks. Furthermore, we believe that achieving superior robustness in self-supervised learning is a crucial research path towards achieving authentic robustness in representation. We hope that our work will inspire more research aimed at achieving generalizable robustness in unseen domains and datasets by leveraging the potential of various self-supervised frameworks.

