# OpenReview forum: "Effective Targeted Attacks for Adversarial Self-Supervised Learning"
_NeurIPS.cc/2023/Conference — NeurIPS 2023 poster_

### Official Review · Reviewer_Uc5D · 2023-07-05

**Soundness:** 2 fair
**Presentation:** 4 excellent
**Contribution:** 3 good
**Rating:** 7
**Confidence:** 5

**Summary:**

This paper highlights that untargeted attacks for adversarial self-supervised learning result in poor downstream robustness. Instead, they suggest utilizing targeted attacks and propose a scoring method for selecting the target samples. They show that this approach results in significant robustness improvements over the untargeted attacks.


**Strengths:**

1. The paper highlights a major issue with adversarial self-supervised learning for non-contrastive models, that the untargeted attacks typically used fail to deliver good downstream robustness.
2. The proposed method shows significant robustness improvement when compared to positive-only untargeted attack.
3. The paper is well-written and the definitions and argumentation are easy to follow.

**Weaknesses:**

1. The proposed method, TARO, has two components but their individual contributions are not studied. TARO improves over the positive-only unsupervised attack by i. changing it to a targeted attack and ii. by selecting the target using a score function. However, there are indications in the prior work and in the results in this paper that the majority of the improvement comes from i. and not ii. For example, (Petrov and Kiwatkowska, 2022) also compare targeted and untargeted adversarial self supervision and show that targeted attacks across the whole batch result in more certifiably robust models than the untargeted ones.  The fact that the improvements in Table 4 in this paper are significantly lower than the improvements in Table 3 is further evidence that it might be i. that drives the improvement rather than ii. Hence, the authors should consider adding an additional baseline to all their results where instead of using their score function, they select a random sample from the batch. This would also be computationally easier, as the paper reports a 5% overhead for the target selection.
2. The paper does not address the fact that the training loss and the loss used for computing the adversarial samples need not be the same. In fact, a number of prior works have this precise setting, e.g., (Alayrac et al., 2019), (Nguyen et al., 2022). Hence, it is not particularly novel that the loss used for training a positive-only model can be different from the one used for the attacks.
3. Only RoCL and ACL are compared against the novel method, but a number of other unsupervised adversarial training methods have been proposed as listed above. In order to claim that the target selection via a score function is better than them, the authors should compare with a larger range of methods. In my view, it is especially important to compare against methods that generate adversarial examples using the whole batch, rather than a single target as these methods might be stronger than the proposed TARO. Some examples of such prior work are (Ho and Vasconcelos, 2020), (Fan et al., 2021), (Petrov and Kwiatkowska, 2022) and the Adversarial-to-Adversarial and Dual Stream methods by (Jiang et al., 2020).
4. In the “Visualization of embedding space” paragraph and Fig. 3, it is actually not that clear whether the targeted attacks generate more samples on the boundary than the untargeted attacks. Visually the two plots seem quite similar.

References:

Jean-Baptiste Alayrac, Jonathan Uesato, Po-Sen Huang, Alhussein Fawzi, Robert Stanforth, and Pushmeet Kohli. Are labels required for improving adversarial robustness?, 2019

Chih-Hui Ho and Nuno Vasconcelos. Contrastive learning with adversarial examples, 2020

Ziyu Jiang, Tianlong Chen, Ting Chen, and Zhangyang Wang. Robust pre-training by adversarial contrastive learning, 2020

Lijie Fan, Sijia Liu, Pin-Yu Chen, Gaoyuan Zhang, and Chuang Gan. When does contrastive learning preserve adversarial robustness from pretraining to finetuning?, 2021

A. Tuan Nguyen, Ser Nam Lim, and Philip Torr. Task-agnostic robust representation learning, 2022

Aleksandar Petrov and Marta Kwiatkowska. Robustness of Unsupervised Representation Learning without Labels, 2022

**Questions:**

1. I did not understand the “Analysis of the selected target”. What is the AT model used? How was it trained? And why was an AT model preferred to a standardly trained one? It is also not particularly clear what Fig. 2a shows. Perhaps more clarity in the writing could help.
2. The notation in the “Similarity and entropy-based target selection for targeted attack” paragraph is confusing. $p’$ and $p$ are used in the equations but $p_i$ is defined below. Is $S_\text{entropy}$ a vector or a scalar? If $p$ is a vector of logits then isn’t $S_\text{entropy}$ a vector?

Typo:
- The sentence starting on line 361, an unnecessary “are” and “belonging” should be “belong”.

**Limitations:**

As mentioned in Weaknesses.

---

> ### Author Rebuttal · Authors · 2023-08-08
>
> **Weakness 1:** Individual contributions of two components (i. changing it to a targeted attack and ii. selecting the target using a score function.) are not studied. Lower improvements in Table 4 compared to Table 3 suggest that [i] might be driving the improvement, rather than [ii].
>
> **Response:** **There seems to be a critical misunderstanding of our work.** Our approach consists of **two interrelated components**, not individual ones: [i] changing to a targeted attack and [ii] selecting the target using a score function. It's important to note that component [ii] is not separate but rather boosts the performance of component [i] (Line 245-251).
> * [i] is proposed based on the theoretical motivation of Theorem 3.2, which suggests that targeted attacks enlarge the scale of attacks (Line 230). Empirically, we demonstrate the effect of the first component in Table 2.
> * [ii] is introduced to find a more effective target rather than random instances, enhancing the robustness of the self-supervised models. As shown in Table 3, the model can achieve improved robustness due to selecting a more effective target for the targeted attack in Equation 9 (Line 227). We summarize the tables in the below.
> |Method|Component|Clean|PGD|
> |-|-|-|-|
> ||-|71.78|32.28|
> |SimSiam|[i]|73.25|42.85|
> ||[i]+[ii]|**74.87**|**44.71**|
> * This improvement also holds true in contrastive-based SSL, such as RoCL and ACL, as indicated in the following table.
> * Specifically, random targeted attacks boost the performance of the original RoCL and ACL. When we employ the score function to select a more effective target for the targeted attack, the model can achieve even better robustness.
> |Method|Attack|Target Selection|Clean|PGD|
> |-|-|-|-|-|
> |RoCL|Untargeted|-|78.14|42.89|
> ||Targeted|Random|79.26|43.45|
> |||Score function|80.06|45.37|
> |ACL|Untargeted|-|79.96|39.37|
> ||Targeted|Random|73.25|42.85|
> |||Score function|78.45|39.71|
> * The reason that the improvements in Table 4 are lower than those in Table 3 is **due to the different frameworks employed.**
> * Table 3, based on positive-pair only SSL frameworks like BYOL and SimSiam, contrasts with Table 4, which shows results from a contrastive learning framework like SimCLR. BYOL and SimSiam benefit more from TARO due to their reliance on positive pairs (Line 312-316).
> * The contrastive-based approach SimCLR diminishes TARO's effect by using both positive and negative pairs in representation learning, reducing the positive pair effect to 1/batch size in loss objectives (Line 321-323).
>
> ---
> **Weakness 2:** The paper does not address the fact that the training loss and the loss used for computing the adversarial samples need not be the same.
>
> **Response:** This is a critical misunderstanding of our contributions. Our primary contribution is not proposing a different attack loss compared to the training loss to gain robustness.
> * Instead, our novel contribution is that we first propose a targeted attack between the positive pair for adversarial self-supervised learning. This proposal is based on theoretical motivation (Theorem 3.2), where a targeted attack can generate stronger adversaries during adversarial representation learning (Table 2).
> * We summarize the tables in the below.
> |SSL|Attack Type|Clean|PGD|
> |-|-|-|-|
> |SimSiam|Untargeted|66.36|36.53|
> ||Targeted|77.08 **(+10.02%)**|47.58 **(+11.05%)**|
> * We propose to **switch from an untargeted attack to a targeted attack** between the original image and the selected target image, i.e. positive-pair images, as outlined in the following equation.
> > Untargeted attack
> $\delta^{t+1}=\Pi_{B(\epsilon)}(\delta^t+\alpha sign(\nabla_{\delta}L(t1(x_i), t2(x_i)))$ \
> > Targeted attack
> $\delta^{t+1}=\Pi_{B(\epsilon)}(\delta^t+\alpha sign(\nabla_{\delta}-L(t1(x_i), t2(x_j)))$
>
> ---
> **Weakness 3:** Limited comparisons.
>
> **Response:** Thank you for the related work suggestions; they will be included in our revision. Since our method suits any adversarial SSL framework using positive-pair images, we applied the TARO method to recent work (DynACL) and confirmed TARO's effectiveness.
> * Please note that since [1] did not provide any official code or checkpoint, we were unable to evaluate this approach against our evaluations. However, we will discuss it in the related work section.
> |Method|Clean|PGD|
> |-|-|-|
> |DynACL|78.56|46.10|
> |DynACL+TARO|**78.83**|**46.79**|
> ---
> **Weakness 4:** It is not that clear whether the targeted attacks generate more samples on the boundary than the untargeted attacks.
>
> **Response:** (Figure also can be found in the PDF file) As depicted in Figure 3 (a), some of the dark blue square dots are positioned near the yellow and orange clusters, but approximately **half of the instances are found within the blue cluster.** These particular instances may not effectively contribute to improving robustness. However, in contrast, Figure 3 (b) shows that **most of the dark blue triangles are aligned along the outline** of the blue cluster.
>
> ---
> **Q1. About “Analysis of the selected target”.**  \
> **R.** We train the AT model, a supervised adversarial training model, with [1]. To identify more attackable classes in robust representation, we use the AT model instead of a standard one. In Figure 2(a), we find that the "ship" class is most susceptible to "plane" image attacks. Our analysis in Figure 2(b) shows that our target selection algorithm often chooses "ship" images as targets for targeted attacks, in line with our aim to create stronger attack images.
> We will revise the paragraph more clearly.
> [1] Towards Deep Learning Models Resistant to Adversarial Attacks
>
> **Q2. About Equation $S_{entropy}$.** \
> **R.** Sorry for the confusion; the symbol $pi$ is a typo, and it should be $p$, where $p=h\circ g\circ f(x)$. Here, p is a vector of logits, and we calculate the entropy which is scalar by summing up the values along the corresponding dimension as follows:
> $S_{\text{entropy}}(x,x') = \sum \frac{p'}{\tau} \log(\frac{p'}{\tau})$

---

> > ### Comment · Reviewer_Uc5D · 2023-08-14
> >
> > Thank you for your detailed response and for providing the additional results. They do indeed show that the target selection increases the resulting robustness a bit over using random selection. I'd recommend adding these results to the manuscript.
> >
> > I have one outstanding question about __W1__ though:
> >
> > You say _Specifically, random targeted attacks boost the performance of the original RoCL and ACL. When we employ the score function to select a more effective target for the targeted attack, the model can achieve even better robustness._ but that doesn't seem to be true for ACL. The robust accuracy is lower there for the score function compared to the random selection. How do you reconcile this?

---

> > > ### Author Response · Authors · 2023-08-14
> > > **Apologize for the typo**
> > >
> > > Thank you for your comments.
> > >
> > > We apologize for the confusion. There appears to be a typo regarding the performance of 73.25/42.85. This performance actually corresponds to the random selection performance of SimSiam as referenced in the first table (second row).
> > >
> > > The correct performance is **78.31/39.74**.
> > >
> > > While the performance gain from the contrastive-based approach is not as significant as the positive-pair SSL framework, it's worth noting that the contrastive-based approach, SimCLR, mitigates TARO's effect by utilizing both positive and negative pairs in representation learning. This reduces the positive pair effect to 1/batch size in loss objectives (see Line 321-323).
> > >
> > > We appreciate your feedback. To provide clarity, we will revise the table in our initial response as soon as you find our corrections.
> > > Moreover, we will certainly adding these results in the manuscripts.
> > >
> > > Best regards,  \
> > > Author

---

> > > > ### Comment · Reviewer_Uc5D · 2023-08-14
> > > >
> > > > Thank you for the clarification. I think that answers my questions and concerns. I would strongly recommend you put the random target baseline in your camera ready and clearly explain under which conditions your method helps and when not. The fact that there is a significant improvement only for positive-pair SSL is somewhat of a limitation, but I think that, as long as it is properly discussed and disclosed (maybe also mention in a "Limitations" section), it does not negate the utility of your work. I have therefore increased my score.

---

> > > > > ### Author Response · Authors · 2023-08-14
> > > > >
> > > > > Dear Reviewer,
> > > > >
> > > > > We sincerely thank you for your feedback. Your positive comments and detailed concerns have greatly enhanced the quality of our work.
> > > > >
> > > > > We will clearly describe our methodology related to the random target baseline and also include the experimental results in the manuscript. We deeply appreciate the time and effort you've dedicated to reviewing our work.
> > > > >
> > > > > Best regards, \
> > > > > Author

---

### Official Review · Reviewer_zysz · 2023-07-06

**Soundness:** 3 good
**Presentation:** 3 good
**Contribution:** 3 good
**Rating:** 7
**Confidence:** 4

**Summary:**

This paper proposes a new adversarial attack on self-supervised learning methods which is called TARO. The method performs an attack "on the positive-pair that perturb the given instance toward the most confusing yet similar latent space, based on entropy and similarity of the latent vectors." Authors show that this attack can be used to improve adversarial robustness of the underlying semi-supervised learning framework.

**Strengths:**

- The experimental evaluation shows that TARO can be used to improve adversarial robustness of positive-pair-only self-supervised learning approaches (SimSiam and BYOL).
- The paper is mostly well-written and comprehensible. It also gives a useful overview of the necessary backgrounds.

**Weaknesses:**

- Mathematical notation can be improved at some points:
	1. In Section 3.1 the authors state that the dataset $\mathcal{D} = \\{X, Y\\}$ is a set of the training data and the corresponding labels. This does not make sense: $\mathcal{D}$ has to be a tuple and X and Y also need to be ordered sets.
	2. For me it is not clear what is meant by ${z_{pos,neg}}$ in Equation 2. The used notation suggests that it is a set containing a single element. However, that does not make sense as the sum in the denominator should go over all positive and negative samples.
	3. In Equation 3 it is not specified at all over which entries the sum is calculated.
- The number of evaluated attacks is too small. The authors only compare against PGD and Autoattack. It might be interesting to compare against other attacks, e.g. Carlini-Wagner or black-box attacks.
- Theorem 3.2 states that "...adversarial perturbations are *likely* to be larger than...". However, the next sentence states that this is always the case. I suggest that the authors reformulate that sentence.
- Figure 1 is difficult to understand. I had to read some more sections first to understand that the positive-pair only attack is proposed in the paper. Maybe, the term should be mentioned earlier (perhaps even in the abstract.)

Minor comment:
- Using verbatim in formulars is quite uncommon. I suggest the authors use \text{}, but this is a matter of taste.
- "positive-pair only" vs. "positive-pair-only"

Overall, I see merit in this submission and willing to recommend acceptance after the weaknesses mentioned in my review were addressed.

**Questions:**

- How improves adversarial training using TARO robustness against other attacks, e.g. Carlini-Wagner or black-box attacks?
- Are there any other SSL methods besides BYOL and SimSiam which could benefit from TARO?

**Limitations:**

Limitations are not discussed in the paper.

---

> ### Author Rebuttal · Authors · 2023-08-07
>
> Thank you for your positive comments on our work, that our work **highlights the improvement of robustness** in the positive-pair-only self-supervised learning approach, **acknowledges the well-written and comprehensive** paper, and **has merits**.
>
> -----
> In the following response, we have done our best to resolve all the concerns that you raised in the weakness section. Please find our detailed response below, and if there are further concerns about our work, do not hesitate to share your comments. We would be delighted to address any additional questions or concerns you may have.\
> Thank you again for your valuable comments and thoughtful review.
>
> ---
> **Weakness 1.** Mathematical notation should be revised.
> * We will revise mathematical notation as follows. Furthermore, we will also go over other parts to improve the clarity of the mathematical notations.
> 1) In Section 3.1 the authors state that the dataset D={X,Y} is a set of the training data and the corresponding labels.
>     * We will revise the definition of dataset as follows: dataset $D=${$(x_i,y_i)$} where $x_i \in R^D$ are input images and $y_i \in R^N$ are their corresponding labels from the N classes.
> 2) For me it is not clear what is meant by zpos,neg in Equation 2.
>     * It is a typo, and we apologize for the confusion. {$z_{\text{pos}}$} and {$z_{\text{neg}}$} are the sets of the latent vector \(z\) of positive-pair and negative pairs, respectively. We will revise Equation 2 as follows:
> $L_{nt-xent} (x,\\{ x_{pos}\\}, \\{x_{neg}\\}) := -log \frac{\sum{z_p\in\\{z_{pos}\\}} (sim(z, z_p)/\tau) } {\sum{z_p \in\\{z_{pos}\\}} (sim(z, z_p)/\tau)+\sum{z_n \in \\{z_{neg}\\}} (sim(z, z_n)/\tau)}$
> 3) In Equation 3 it is not specified at all over which entries the sum is calculated.
>     * Equation 3 represents the objective of SimSiam [1], a positive-pair only SSL framework. The loss function calculates the cosine similarity between the positive pairs, which are originally the same instance but augmented differently. For clarification, we will revise the Equation as follows:
> $L_{ss}(x, x_{pos})$=$L_{t_1(x), t_2(x)}=-\frac{1}{2}\frac{p_1}{||p_1||_2}\frac{z_2}{||z_2||_2}-\frac{1}{2}\frac{p_2}{||p_2||_2}\frac{z_1}{||z_1||_2}$
> where $p_i =h \circ z_i$ and $z_i=g\circ f(t_i(x))$.
>
> [1] Exploring simple siamese representation learning., CVPR 2021
>
> ---
> **Weakness 2./ Q1.** Evaluation against other attacks is needed.
> * Thank you for your comments. Following your suggestions, we tested our approach against the Carlini-Wagner (CW) attack [1], black-box attacks, i.e., Pixle [2], and Patch-attack, i.e., PIFGSM [3], as shown in the following table. Since our approach already showed improved performance against Autoattack, which includes black-box Square attacks, our approach is able to consistently demonstrates enhanced robustness against both the CW attack and AT black-box attacks.
> ||PGD|CW [1]|Pixle [2]|PIFGSM [3]|
> |-|-|-|-|-|
> |RoCL|42.89|76.45|67.32|43.23|
> |RoCL+TARO|45.37|72.75|68.40|44.56|
> |SimSiam|32.28|68.14|54.56|28.31|
> |SimSiam+TARO|44.97 |73.87|67.22|46.37|
>
> [1] Towards Evaluating the Robustness of Neural Networks, 2017 \
> [2] Pixle: a fast and effective black-box attack based on rearranging pixels, ECCV2020 \
> [3] Patch-wise Attack for Fooling Deep Neural Network, 2022
>
> ---
> **Weakness 3.** Theorem 3.2 should be reformulated.
> * Thank you for your comments. We will reformulate Theorem 3.2 as follows to make the theorem more clear.
> > Theorem 3.2 **[Perturbation range of targeted attack]** Given a model trained under the $L_{\text{targeted-attack}}$ loss, the adversarial perturbations $\delta_{\text{targeted-attack}}$ are larger than the adversarial perturbations $\delta_{\text{ss}}$ from a model trained under the $L_{ss}$. Formally, $|\delta_{\text{targeted-attack}}|$ $>\|\delta_{\text{ss}}\|_\infty$.
>
> ----
> **Weakness 4.** Figure 1 is difficult to understand and positive-pair only attacks should be explained earlier.
> * We will revise the abstract to include mention of the positive-pair-only attack for better readability. Furthermore, we have also revised Figure 1, providing clearer descriptions for the captions in the PDF.
>
> ----
> **Minor comment:**
>     We will revise verbatim in formulas and use \text{}. Moreover, we will use the consistent term “positive-pair only” in all manuscripts for better consensus of the term. Thanks for the comments!
>
> ---
> **Question.** Are there any other SSL methods besides BYOL and SimSiam which could benefit from TARO?
> * Our TARO can leverage any kind of SSL method that employs positive pairs to learn visual representation. Therefore, methods like BYOL and SimSiam, which solely rely on positive-pair-only frameworks, show larger benefits from TARO (Table 3). Furthermore, the contrastive-based approach SimCLR, which also employs positive pairs in representation learning along with negative pairs, also benefits from TARO as shown in Table 4.
> * Based on your question, we further examine the applicability of TARO by applying it to adversarial self-supervised learning in an additional self-supervised learning framework: Barlow Twins. Barlow Twins is a framework that learns visual representation using positive pairs and redundancy regularization.
> * Since there are no adversarial baselines based on Barlow Twins [2], these results may underfit due to a simple combination of [1] and Barlow Twins [2]. However, when we apply TARO to the adversarial Barlow Twins using the same hyper-parameters and the same settings, **it shows much better clean performance and robustness, as demonstrated in the following table**. This further demonstrates the applicability of our TARO on positive-pair of SSL frameworks.
> |Method|Clean|AutoAttack Robustness|
> |-|-|-|
> |[1]+Barlow Twins [2] | 61.94|11.71|
> |[1]+Barlow Twins [2] + TARO|75.37 **(+13.43)**|26.43 **(+14.72)**|
>
> [1] Kim et al., Adversarial Self-Supervised Contrastive Learning, NeurIPS 2020 \
> [2] Zbontar et al., Barlow Twins: Self-Supervised Learning via Redundancy Reduction, ICML 2021

---

> > ### Comment · Reviewer_zysz · 2023-08-14
> >
> > I would like to thank the authors for their rebuttal and appreciate the effort put into the additional experiments. Overall, most of my concerns have been adequately addressed which is why I increased my rating to 7 and presentation score to 3. However, in my optinion the notation used for the $L_\text{nt-xent}$ loss is still misleading as $\\{z_\text{pos}\\}$ refers to a set containing a single element $z_\text{pos}$ which is not what the authors intent to say. I urge the authors to properly define the set of positive and negative examples.

---

> > > ### Author Response · Authors · 2023-08-14
> > >
> > > Dear Reviewer,
> > >
> > > We sincerely thank you for your feedback. Your positive comments and detailed concerns have significantly helped improve our work. In light of your feedback, we will clarify z_pos and z_neg to prevent any misunderstandings for the readers. We greatly appreciate the time and effort you've dedicated to reviewing our work, and your insights have been invaluable.
> > >
> > > Best regards,   \
> > > Author

---

### Official Review · Reviewer_A38P · 2023-07-07

**Soundness:** 3 good
**Presentation:** 3 good
**Contribution:** 3 good
**Rating:** 6
**Confidence:** 3

**Summary:**

The paper proposed the targeted attack for adversarial self-supervised learning to solve the suboptimal learning issue in positive pair-only self-supervised learning. To improve the robustness of adversarial self-supervised learning (SSL), the author leverages the targeted selection mechanism based on the score function and generate the samples by selected targets for adversarial training. The paper empirically validates that target attacks can improve the adversarial robustness of several previous adversarial SSL methods including BYORL and SimSiam.

**Strengths:**

The paper is well-organized and easy to follow. The proposed idea is clear and has good motivation.

**Weaknesses:**

For the transferable robustness part, the author mentioned that TARO can transfer robustness from one task to another. In my opinion, CIFAR100 and CIFAR10 share a similar domain and is not trivial to transfer the robustness. Although the authors evaluate on AutoAttack, which is a well-known strongest attack so far, one thing that I might be curious about is the transferability from the PGD attack (known attack) to other kinds of unknown attacks such as the Patch attack, CW_l2.

**Questions:**

Can the proposed method be applied to the transformer-based model, such as ViT?

**Limitations:**

yes, the authors adequately addressed the limitation

---

> ### Author Rebuttal · Authors · 2023-08-07
>
> Thank you for your positive comments on our work, which highlight good motivation clear idea with well-organized writing.
>
> ---
> In the following response, we have done our best to resolve all the concerns that you raised regarding our work. Please find our detailed response below, and if there are further concerns about our work, do not hesitate to share your comments. We would be delighted to address any additional questions or concerns you may have during the discussion phase.
>
> Thank you again for your valuable comments and thoughtful review.
>
> ---
> **Weakness 1.** CIFAR100 and CIFAR10 seem not trivial to transfer the robustness.
> * As you commented we also tested on the different domains transferring from CIFAR10 to STL10, the robustness in the following table. Consistently, our approach shows better-transferring robustness in non-trivial transfer tasks.
> * However, please note that transferring the robustness is one of the challenging tasks to achieve even from a similar domain from CIFAR100 to CIFAR10 [1]. Therefore, many previous works have conducted experiments on CIFAR100 to CIFAR10, and we followed the same setting as [1,2,3].
>
> | CIFAR10  | target dataset| Clean | PGD $\ell_{\infty}$ |
> |--|:-------:|-----------|-----------|
> |RoCL|STL10|63.85|32.75|
> |RoCL + TARO|STL10|**67.30**|**32.90**|
> |SimSiam|STL10|30.22|12.66|
> |SimSiam + TARO |STL10|**54.45**|**33.44**|
>
>
> [1] Shafahi et al., Adversarially Robust Transfer Learning, ICLR 2020 \
> [2] Kim et al., Adversarial Self-Supervised Contrastive Learning, NeurIPS 2020 \
> [3] Fan et al., When does Contrastive Learning Preserve Adversarial Robustness from Pretraining to Finetuning?, NeurIPS 2021
>
> ----
> **Weakness 2.** Attack transferability to CW or Patch attack.
> * Following your suggestions, we evaluated our approach against Carlini-Wagner (CW) attack [1], black-box attacks, i.e., OnePixel [2], and Patch-attack, i.e., PIFGSM [3], to demonstrate transferability of the attack types.
> * As shown in the following Table, our works are able to demonstrate the transferable robustness against diverse attacks in both contrastive-based TARO and positive-pair only based TARO.
> |Type|Method|PGD|CW [1]|OnePixel [2]|PIFGSM [3]|Average|
> |-|-|-|-|-|-|-|
> |Contrastive-based SSL|RoCL|42.89|76.45|67.32|43.23|57.47|
> |Contrastive-based SSL|RoCL+TARO|45.37|75.38|68.40|44.56|**58.43**|
> |Positive-pair only SSL|SimSiam|32.28|68.14|54.56|28.31|45.82|
> |Positive-pair only SSL|SimSiam+TARO|44.97|73.87|67.22|46.37|**58.11**|
>
> [1] Towards Evaluating the Robustness of Neural Networks, 2017  \
> [2] One pixel attack for fooling deep neural networks, 2018 \
> [3] Patch-wise Attack for Fooling Deep Neural Network, 2022
>
> ----------
>
> **Question. Can TARO be applied to ViT?**
> * Our work is a model-agnostic approach so that TARO can be applied to a transformer-based model.
> * We are currently running experiments on ViT with TARO. However, due to the computational costs, it will take a few days to obtain the full set of results, and we will soon update our table as soon as they are out.

---

> > ### Comment · Reviewer_A38P · 2023-08-18
> >
> > Thanks for the authors' response and effort. The authors' response has addressed most of my concerns. I would expect the experiment of ViT with TARO can also get better performance. Thus, I will keep my rating.

---

> > > ### Author Response · Authors · 2023-08-21
> > >
> > > Dear Reviewer,
> > >
> > > We apologize for the delay in our response. We encountered challenges because there is no baseline research that has explored either self-supervised learning with a small dataset or adversarial training on ViT. We had to identify an approach that fits within our computational budget while thoroughly evaluating the effectiveness of our TARO.
> > >
> > > Kindly understand that due to computational constraints, our initial efforts did not involve extensive hyper-parameter tuning. To explain the settings we implemented, we adopted the self-supervised ViT approach as described in [1]. Building on [1] which is pretrained on ImageNet1K, we further trained the model for 15 epochs by integrating adversarial self-supervised training using both untargeted and targeted attacks on CIFAR10.
> > >
> > > Here's a summarized table of our results:
> > > |Model|Clean Accuracy|Robustness|
> > > |-----|--------------|---------|
> > > |ViT [1]+untargeted attack|66.63%|56.36%|
> > > |ViT [1]+TARO|74.01%|67.23%|
> > >
> > > As illustrated in the table, **TARO has also demonstrated effectiveness when applied to ViT** since TARO is a model-agnostic approach.
> > >
> > > [1] Emerging Properties in Self-Supervised Vision Transformers
> > >
> > > Best, \
> > > Author

---

### Official Review · Reviewer_aDD2 · 2023-07-12

**Soundness:** 2 fair
**Presentation:** 3 good
**Contribution:** 2 fair
**Rating:** 4
**Confidence:** 5

**Summary:**

This paper investigates the problem of unsupervised adversarial training. It claims that previous non-contrastive, positive-only SSL frameworks suffer from ineffective learning with untargeted adversarial samples. Based on this, this paper proposes a new TARO paradigm, which conducts targeted attacks to select the most confusing but similar samples for guiding the gradients toward a desired direction. Experiments show some great results.

**Strengths:**

1. The motivation of this paper is clear, which addresses the limitation so existing positive-only SSL frameworks.
2. The paper is well-organized and easy to read.
3. Supplementary file is provided.

**Weaknesses:**

1. The novelty is incremental. Actually, the technical design of targeted adversarial SSL is too straightforward. Although many theorems are provided in the paper to support it, I don't think they are critical to the final model design.

2. Missing many related works. The references are out-of-data, they are all published before 2022. There are many recent papers of this topic, the authors should carefully revisit them and discuss with them.

3. The experiments are unconvincing. The baselines are old, the authors should conduct experiments with latest baselines. Moreover, the ablation study is also insufficient.

**Questions:**

See above

---

> ### Author Rebuttal · Authors · 2023-08-07
>
> Thank you for your positive comments on our contributions, which highlight clear motivation and address the limitations of existing positive-pair-only SSL frameworks in robustness.
>
> ----
>
> In the following response, we have done our best to resolve all the concerns that you raised regarding our work. Please find our detailed response below, and if there are further concerns about our work, do not hesitate to share your comments. We would be delighted to address any additional questions or concerns you may have.
>
> Thank you again for your valuable comments and thoughtful review.
>
> ----
>
> **Weakness 1.** The novelty is incremental.
> * We appreciate the feedback, but it appears there may be a fundamental misunderstanding of our work that requires clarification.
> * In Table 2 of our manuscript (also summarized in the following table), we observe that robustness in positive-pair only SSL is extremely vulnerable compared to success in adversarial contrastive-based SSL. The vulnerability in positive-pair only SSL can be attributed to the restricted range of perturbation, in contrast to that generated by contrastive-based SSL, as elucidated in Theorem 3.1. Based on this observation, we propose a simple yet effective approach wherein a targeted attack can be employed to augment the perturbation range, as outlined in Theorem 3.2.
> * We would like to underscore that, to the best of our knowledge, **the utilization of targeted attacks between two instances has not been previously explored** in the field of adversarial SSL.
> * **Our approach is founded on theoretical motivation** that suggests a targeted attack amplifies the range of attack strength in positive-pair only SSL (as detailed in Section 3, Lines 185-214). This method involves a methodical design that conducts targeted attacks between positive-pair instances to enhance the robustness of adversarial SSL.
> * Our novel score function proposal involves **selecting more strategic targets, rather than random instances, to increase robustness** in adversarial SSL, building on prior studies [1,2,3].
> * Our empirical findings ensure the effectiveness of our score function design. As shown in the corresponding table, our score function outperforms random selection in terms of robustness, thereby substantiating the merits of our method.
> |Type|Method|Target selection|Clean|Robustness|
> |-|-|-|-|-|
> |Positive-pair only SSL|SimSiam|-|71.78|32.28|
> |Positive-pair only SSL|SimSiam+TARO|Random|73.25|42.85|
> |Positive-pair only SSL|SimSiam+TARO|Score function|74.87 **(+3.09)**|44.71 **(+12.43)**|
>
> * Our targeted adversarial SSL can be adapted to any kind of adversarial SSL framework including contrastive-based SSL and non-contrastive-based SSL, i.e., positive-pair-only SSL, which is especially effective in positive-pair-only SSL (Table 3). The summarized table is as shown follows:
> |Type|Method|Clean|Robustness|
> |-|-|-|-|
> |Contrastive-based SSL|RoCL|78.14|42.89|
> |Contrastive-based SSL|RoCL+TARO|80.06 **(+1.92)** |45.37 **(+2.48)**|
> |Positive-pair only SSL|SimSiam|71.78|32.28|
> |Positive-pair only SSL|SimSiam+TARO|74.87 **(+3.09)**|44.71 **(+12.43)**|
>
> [1] Mma training: Di- rect input space margin maximization through adversarial training. ICLR 2020 \
> [2] Evaluating the robustness of geometry-aware instance-reweighted adversarial training. ICLR 2021 \
> [3] Rethinking the entropy of instance in adversarial training., SaTML 2023
>
> -----
> **Weakness 2**. Many recent related works are missing.
> * Thanks to your comments, we will add recent references [4,5,6] in our paper including the publication in 2023.
> * Since our approach can be applied to various kinds of adversarial SSL, including the contrastive-based SSL approach, we applied TARO to DynACL [6]. As shown in the response to Weakness 3 below, experimental results show that our approach is also effective in [6]. If you have additional publications that need to be revisited in our paper, please let us know so we can include them in our comparison.
>
> -----
> **Weakness 3.** Experiments are not convincing due to old baselines and insufficient ablation study.
> * As we mentioned in the previous response, we additionally conduct the experiment on top of the most recent work, i.e., DynACL'23 [6].
> * As illustrated in the table below, TARO is also effective with DynACL, a recent variation of ACL that schedules the data augmentation strength during pretraining. Since TARO proposes to adjust the attack to be stronger than the original ACL attack, our approach is applicable and can demonstrate effectiveness with this recent variation of ACL, specifically DynACL.
> | | Clean | PGD| AutoAttack |
> |-|-|-|-|
> |DynACL|78.56|46.10|43.31|
> |DynACL+ TARO|**78.83**|**46.79**|**43.53**|
>
> * Furthermore, we conducted an ablation study on our score function to confirm the effectiveness of our selection algorithm within both the contrastive learning framework and the positive-pair only framework. Compared to random selection, our score function exhibits superior robustness in adversarial SSL, thereby underscoring the advantages of our approach.
> * Please let us know if any further ablation studies are needed to understand our approach.
> ||Target Selection|Clean|PGD| AutoAttack |
> |-|:-:|-|-|-|
> |SimSiam|-|71.78|32.28|24.41|
> |SimSiam+TARO|Random|73.25|42.85|34.72|
> |SimSiam+TARO|Score function|**74.87**|**44.71**|**36.39**|
> |RoCL|-|78.14|42.89|27.19|
> |RoCL+TARO|Random|79.26|43.45|27.24|
> |RoCL+TARO|Score function|**80.06**|**45.37**|**27.95**|
>
> [4] Xie et al., Adversarial examples improve image recognition, CVPR'20 \
> [5] Fan et al., When does Contrastive Learning Preserve Adversarial Robustness from Pretraining to Finetuning?, NeurIPS'21 \
> [6] Luo and Wang et al., Rethinking the Effect of Data Augmentation in Adversarial Contrastive Learning, ICLR'23

---

> > ### Comment · Reviewer_aDD2 · 2023-08-14
> >
> > Thanks for the authors' clarification. The authors' response has addressed most of my concerns. However, I still think the contributions of this paper are not enough to meet the standards of the Neurips conference. Therefore, I will keep my rating.

---

> > > ### Author Response · Authors · 2023-08-14
> > >
> > > Thank you for your comments and feedback. And we are glad that we resolve most of your initial concerns.
> > >
> > > However, we kindly seek clarity regarding the criteria by which our contributions did not meet the standards of the NeurIPS conference, especially since we have addressed most of the concerns you raised.
> > >
> > > Given our theoretical motivations and the results of our empirical experiments, we are confident that our work offers meaningful insights to the adversarial self-supervised learning community.
> > >
> > > Could you please elaborate on any remaining concerns about our contribution? We are willing to address them if possible.
> > >
> > > Best, \
> > > Author

---

### Author Rebuttal · Authors · 2023-08-09

Dear Reviewers,

We would like to thank you for the time and effort you've invested in reviewing our paper, and for the constructive feedback you have provided. During the initial response period, we did our best to address all the concerns you raised and to improve our paper according to your insights. We have responded to each of your individual comments. Furthermore, we have additional discussions and experimental results from various perspectives, in line with your suggestions. A brief summary of our responses is provided below for your convenience. We hope that our revisions have resolved your concerns, and kindly request you to consider reflecting these changes in your updated review scores.

---
* Reviewer aDD2: Clarified our contributions and approach
* Reviewer aDD2: Added experiments based on recent work (DynACL’23)
* Reviewer aDD2: Conducted ablation experiments on the score function

---
* Reviewer A38P: Added transfer tasks to the STL10 dataset
* Reviewer A38P: Introduced additional adversarial evaluations against CW, Patch attack, and black-box attack

---
* Reviewer zysz: Revised mathematical notations
* Reviewer zysz: Updated Figure 1 and its descriptions (in PDF file)
* Reviewer zysz: Included additional adversarial evaluations against CW, Patch attack, and black-box attack
* Reviewer zysz: Experimented with different types of SSL frameworks, such as Barlow Twins, using our approach

---
* Reviewer Uc5D: Conducted an ablation study on each component (i.e., targeted attack, target selection algorithm)
* Reviewer Uc5D: Analyzed the ablation study of target selection algorithms in contrastive-based SSL frameworks
* Reviewer Uc5D: Added comparison experiments based on recent work (DynACL’23)
* Reviewer Uc5D: Added detailed description of Figure 2, 3 (in PDF file)

---

### Decision · Program_Chairs · 2023-09-21

**Decision:**

Accept (poster)

**Comment:**

The paper received mixed scores before the rebuttal. The reviewers all think the motivation is clear and the paper is well-written. The main concerns from some reviewers are the technique is not sound (the design of TARO) and the empirical studies are not solid (limited methods in comparison, marginal gain, etc.). After the rebuttal, some of the concerns have been addressed. AC read the paper and all the feedback from the authors/reviewers and thinks this is a borderline paper. After a discussion with SAC and PCs, the novelty and the good empirical result of the paper are still appreciated. Therefore, the final decision is yes.